# Understanding the Propagation of Meteorological Drought to Groundwater Drought: A Case Study of the North China Plain

Yuyin Chen [1,2], Yongqiang Zhang [1,*], Jing Tian [1], Zixuan Tang [1,2], Longhao Wang [1,2] and Xuening Yang [1,2]

1   Key Laboratory of Water Cycle and Related Land Surface Processes, Institute of Geographic Sciences and Natural Resources Research, Chinese Academy of Sciences, Beijing 100101, China; tangzx.20s@igsnrr.ac.cn (Z.T.)
2   University of Chinese Academy of Sciences, Beijing 100049, China
*   Correspondence: zhangyq@igsnrr.ac.cn

**Abstract:** As extreme climate events become more common with global warming, groundwater is increasingly vital for combating long-term drought and ensuring socio-economic and ecological stability. Currently, the mechanism of meteorological drought propagation to groundwater drought is still not fully understood. This study focuses on the North China Plain (NCP), utilizing statistical theories, spatiotemporal kriging interpolation, and the Mann–Kendall trend test to examine the spatial and temporal distribution characteristics of groundwater from 2005 to 2021. Based on drought theory, the characteristics and propagation process of drought are further quantified. Key findings reveal the following: (1) Shallow groundwater depths in the NCP follow a zonal pattern from the western mountains to the eastern plains and coastal areas. Over two-thirds of this region showed an increase in groundwater depth at a rate of 0–0.05 m/a; (2) Groundwater drought frequency typically ranges from 3 to 6 times, with an average duration of 10 to 30 months and average severity between 10 and 35; (3) Delayed effects last between 0 to 60 months, with attenuation effects varying from 0 to 3 and prolonged effects extending from 0 to 16. Additionally, delayed effects intensify with increasing time scales, while prolonged effects weaken. Notably, both delayed and prolonged effects in the north of the NCP are more pronounced than in the south of the region. This study quantifies the process by which meteorological drought propagates to groundwater drought, offering a new perspective for understanding the interaction between groundwater and meteorological drought. It holds significant scientific importance for monitoring drought and managing water resources in the context of global climate change.

**Keywords:** groundwater drought; meteorological drought; response relationship; drought propagation



## 1. Introduction

The significant increase in the frequency and duration of global droughts has attracted widespread attention in the fields of meteorology, agriculture, and hydrology [1]. In 1997, the American Meteorological Society defined four main types of droughts: meteorological drought, agricultural drought, hydrological drought, and socio-economic drought. Meteorological drought is characterized by a prolonged period of significantly below-average precipitation, leading to insufficient water supply for the established needs of a region. This type of drought is quantified by comparing the current precipitation levels with historical averages over a given time frame and area. It is the initial manifestation of drought that can eventually lead to other forms of drought if the deficit persists [2]. In recent years, groundwater and ecosystem drought have also been progressively included in the drought assessment framework [3]. Studying the propagation patterns from meteorological droughts to different types of secondary droughts is the key to the scientific response to future extreme climate.

However, the complexity and diversity of droughts make characterization and quantification of relationships extremely challenging. Researchers have measured and compared

drought severity by introducing drought indicators [4–6]. Similar to the classification of drought types, existing drought indicators can be broadly categorized into meteorological drought indicators, hydrological drought indicators, agricultural drought indicators, and socio-economic drought indicators. In addition, groundwater drought indicators have been developed [7]. The spatial and temporal evolution of drought can be quantified from several perspectives by calculating different drought indices. Due to the differences in drought characteristics in different regions, it is often necessary to select the index that best characterizes regional drought based on climatic and watershed characteristics [8]. Meteorological drought assessment indicators are usually categorized into single-factor and multifactor categories [9–15]. A further summary of the current commonly used drought indices is provided in our supplement (Supporting Information Table S3). Among various meteorological drought indicators, the standardized precipitation evapotranspiration index (SPEI) is widely applied. This index, based on the water balance, takes into account the influences of both precipitation and evapotranspiration, providing a comprehensive monitoring ability. The applicability of the SPEI in the North China Plain has been widely recognized [16,17].

The research on the relationship between groundwater and meteorological drought originated in 1977 when researchers observed anomalous declines in groundwater levels associated with regional drought in the United Kingdom, especially in aquifers with lower storage capacity [18]. However, these early studies failed to observe the permanent adverse effects of drought on the aquifer. It was not until 1985 that researchers first included groundwater as one of the parameters for drought monitoring [19]. The concept of groundwater drought was first introduced in 1997 [20] and then clearly defined in 2000 as groundwater level below critical level [21]. By 2003, researchers began using linear reservoir theory to analyze the effects of drought on groundwater recharge and discharge [22]. It was shown that the propagation of drought in the groundwater system is characterized by attenuation and delay. In subsequent studies, various groundwater drought indices have been proposed. For instance, the groundwater resources index (GRI) [23], based on a distributed water balance model, not only allows retrospective analysis of significant historical drought events but can also be employed for predicting summer droughts. However, due to its complex computation and high data requirements, this index has not been widely adopted. The standardized groundwater level index (SGI) [24], estimated using nonparametric normal score transformation of monthly groundwater level data, has gained popularity due to its unique input variables and straightforward computation, facilitating comparisons of groundwater drought across different locations and seasons. The standardized groundwater discharge index (SGDI) [25], estimated based on low-frequency runoff signals, aims to study groundwater drought conditions in regions with scarce well data. Groundwater drought index based on the land surface model (GWI) [26], GRACE (Gravity Recovery and Climate Experiment) groundwater drought index (GGDI) [27], and GRACE-DSI [28] enable real-time monitoring of the spread and severity of groundwater drought on a large scale. However, these indices suffer from coarse resolution and significant uncertainties. The propagation characteristics of meteorological drought to groundwater drought vary significantly depending on geographical location, climatic conditions, land cover, and geological factors. Han et al. [28] revealed that the propagation time of meteorological drought to groundwater drought in the Xijiang River Basin, China, ranged from 8 to 42 months. By analyzing the relationship between SGDI and SPI, Zhu et al. [25] found that groundwater drought in southeastern Australia responded to precipitation drought in more than one year and that there were significant spatial differences in the resistance of the groundwater system during the drought period and its resilience at the end of the drought. According to Zhao et al. [17], drought sensitivity and propagation rate in the NCP show a pattern of high in the north and low in the south.

In groundwater drought studies, although groundwater drought indices based on GRACE and GLDAS (Global Land Data Assimilation System) provide a macroscopic view, due to the low spatial resolution and accompanying uncertainty, these methods have limi-

tations. In prior investigations, there has been a lack of comprehensive understanding of the mechanisms driving the propagation of meteorological–groundwater drought. Particularly, research on meteorological to groundwater drought propagation based on measured data has been relatively limited. There is still a research gap on the spatial distribution characteristics of the cumulative, delayed, attenuation, and prolonged effects of drought propagation. Consequently, there is a pressing need for a more thorough exploration of the essential features of meteorological–groundwater drought propagation to address the existing research gaps.

To improve the precision and reliability of the study, this study combines high-resolution in situ observation data from nearly 1800 wells in the North China Plain to carry out the following work:

(1) Comprehensively analyze the spatial distribution and time-series changes in groundwater in the North China Plain from 2005 to 2021 by applying statistical theory, spatiotemporal kriging interpolation, and Mann–Kendall trend tests;

(2) Calculate the multiscale meteorological drought index (SPEI) and groundwater drought index (SGI) and extract the key features of drought, such as duration, intensity, and frequency, based on the run theory;

(3) Match the two types of drought events and calculate the peak lag time, attenuation scale factor, and extension scale factor of the SPEI and SGI on different time scales to precisely quantify the drought propagation process.

The results contribute to a further understanding of the characteristics and complex propagation relationships between groundwater and meteorological droughts in the NCP.

## 2. Materials and Methods

### 2.1. Study Area

The North China Plain (NCP) is one of the three major plains in China, located at 32~43° N and 110~123° E (Figure 1). Enclosed by mountains to the north, west, and south and bordered by the Bohai and Yellow Seas to the east, the NCP spans five provinces. The NCP is mainly formed by the alluvial deposits of the Yellow River, Huai River, Hai River, and Luan River. The lower reaches of the Yellow River run naturally across the center of the NCP, which is divided into two parts: the Yellow and Huai River Plains in the south and the Hai River Plain in the north. The region's topography is primarily low-lying and flat, with most areas positioned below 50 m above sea level and the eastern coastal plains even lower, beneath 10 m. Sloping gently toward the Bohai Bay, the NCP forms a comprehensive hydrogeological unit stretching from the mountain fronts to the coastline, segmented from west to east into the piedmont alluvial–flood plain, the central alluvial–lacustrine plain, and the eastern alluvial–oceanic plain. Climatically, the area experiences a warm temperate monsoon regime characterized by an average annual precipitation of around 600 mm. This precipitation is unevenly distributed throughout the year, typically lesser in spring and winter, and more abundant in summer and autumn, contributing to the area's frequent spring droughts. In the past 60 years, the temperature increased at a rate of 0.3 °C/10a, and the climate has become warmer and dryer [29]. The main source of groundwater recharge in the North China Plain is seasonal rainfall, and river seepage and mountain lateral runoff contribute a little to groundwater recharge. Therefore, the NCP is an ideal study area for groundwater research. Table 1 provides more information on the study area.

**Table 1.** Basic information on the study area.

| Features | Description |
| --- | --- |
| Provinces and cities | Beijing, Hebei, Tianjin, Henan, Shandong |
| Total covered area | 538,000 km$^2$ |
| Longitude | 110~123° E |
| Latitude | 32~43° N |
| Mean elevation | 312 m |

**Table 1.** *Cont.*

| Features | Description |
|---|---|
| Annual mean precipitation | 340~910 mm |
| Annual mean temperature | 10~15 °C |
| Proportion of land cover | Cropland (85.2%), forest (9.8%), grassland (2.2%), urban areas (2.1%), water bodies (0.6%), bare areas (0.1%) |
| Monitoring well count | 1826 |
| Staple crop | Winter wheat, summer corn, spring corn |
| Area of crops most affected by drought | 4.3 million hectares (2009) |
| Population with drinking water difficulties due to drought | 3.3 million (2010) |
| Large livestock with difficulty in drinking water due to drought | 1.3 million (2006) |

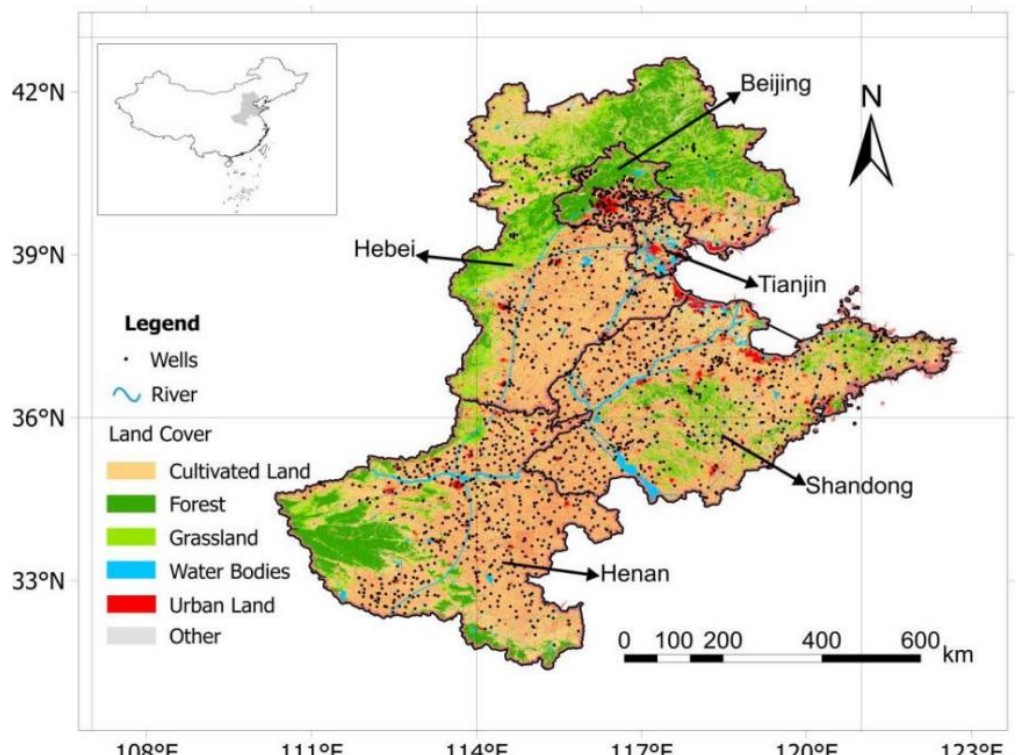

**Figure 1.** Monitoring well distribution and land cover in the study area.

*2.2. Data Source*

2.2.1. In Situ Groundwater Observations

The China Institute of Geo-Environmental Monitoring (CIGEM) has publicly published a total of 17 "China Geo-Environmental Monitoring Groundwater Level Yearbooks" for 2005–2021. The Yearbook contains information on groundwater depth (depth of burial) and basic information on national groundwater monitoring points. The table of basic information includes uniform number, location of monitoring point, longitude, latitude, ground elevation, monitoring depth, hydrogeological unit, type of aquifer, and type of groundwater. It is worth noting that the amount of publicly available groundwater monitoring data has increased significantly after 2018 compared to before 2017. The spatial distribution of monitoring wells is shown in Figure 1.

2.2.2. GLDAS-2.2 Groundwater Storage

NASA's Global Land Data Assimilation System version 2 (GLDAS-2) has three components: GLDAS-2.0, GLDAS-2.1, and GLDAS-2.2. Of these, the GLDAS-2.2 GRACE-DA product is simulated with the Land Information System (LIS) version 7 Catchment-F2.5 in Land Information System (LIS) version 7. The data product contains 24 fields from

1 February 2003 to the present, with "GWS_tavg" representing groundwater storage data in mm. We downloaded the GLDAS-2.2 groundwater storage dataset from Google Earth Engine [30]. These data are used for subsequent comparison with the interpolated results.

### 2.2.3. Meteorological Data

The precipitation, air temperature, and evapotranspiration data were obtained from the "China 1-km resolution monthly precipitation dataset" [31], "China 1-km resolution monthly mean temperature dataset" [32], and the "China 1-km resolution potential evapo-transpiration dataset" [33]. The spatial resolution is 0.0083333° (~1 km), and the data are all in NETCDF format.

### 2.2.4. Land Cover Data

The China Multiperiod Land Use and Land Cover Change Remote Sensing Monitoring Dataset (CNLUCC) is a national-scale 1:10 scale multiperiod land use/land cover thematic database constructed through manual visual interpretation, using Landsat remote sensing images as the main data source. The data adopt a two-tier classification system, with the first tier divided into six categories: cropland, forest land, grassland, water body, built-up area, and unused land, and the second tier further divided into 25 types based on the first-tier types [34].

### *2.3. Methods*

### 2.3.1. Spatiotemporal Kriging Interpolation

Continuous long-series monitoring data are the basis for drought research, but there are often station malfunctions, abandonments, and transfers in the process of conducting in situ groundwater monitoring, resulting in missing and anomalous sequence data. In addition, since the implementation of the comprehensive automation of national groundwater monitoring stations in 2018, the frequency of groundwater level monitoring in time and the density in space have greatly increased. This has further exacerbated the problem of uneven and discontinuous groundwater data distribution, especially around 2018, when the amount of data showed a cliff-like change. To address this problem, one option is to select all the long time series sites for analysis, but this creates a significant waste of existing data resources. Another option is to fill in the gaps through interpolation techniques. It is also possible to simulate groundwater changes through mathematical models based on physical processes [35–37]. In this study, the spatiotemporal kriging (STK) method was used to interpolate and obtain the month-by-month groundwater depth raster data (0.05° × 0.05°) to provide relatively complete and reliable information for the study of the spatiotemporal evolution of groundwater in the NCP. The statistical and Mann–Kendall trend analysis methods used in this study are explained in detail in the supplementary materials (Supporting Information Text S1), and the space–time kriging interpolation principle is mainly introduced in this section. This method has been widely used to model the spatiotemporal processes of temperature [38], wind speed [39], soil water content [40], and groundwater level [41]. STK interpolation takes into account both temporal and spatial correlation information and can realize the mapping from spatiotemporal discrete points to the continuum. The limitation is that the model requires a large amount of calculation, and the choice of spatiotemporal variance function will affect the modeling accuracy. Assuming the existence of $s = (x,y) \in \mathbb{R}^2$, $t \in T$, the following spatiotemporal random variable $Z(s,t)$ can be defined as [42]

$$Z(s,t) = Z(x,y,t). \tag{1}$$

To obtain the values of the spatiotemporal variability function at any spatiotemporal distance, it is necessary to fit a spatiotemporal theoretical variability function model based on the experimental variability function scatter data. Common spatiotemporal covariance models are the metric model, the sum metric model [43,44], the product model [39], the

Cressie–Huang model [45], and the product–sum model [46]. The sum metric model is adopted in this study [40]:

$$C_{st}(h_s, h_t) = C_s(h_s) + C_t(h_t) + C_{joint}\left(\sqrt{|h_s|^2 + (\mathrm{k}\cdot h_t)^2}\right). \tag{2}$$

The sum metric model decomposes the spatiotemporal covariance $C_{st}$ into the sum of three parts: the pure spatial covariance $C_s$, the pure temporal covariance $C_t$, and the spatiotemporal interaction covariance $C_{joint}$. All three independent covariances of groundwater depth in the study area were fitted with a spherical model, which is defined as follows [46]:

$$\mathrm{C(h)} = \begin{cases} 0, & h = 0 \\ C_0 + C\left(\frac{3h}{2a} - \frac{h^3}{2a^3}\right), & 0 < h \le \mathrm{a}, \\ C_0 + C, & h > \mathrm{a} \end{cases} \tag{3}$$

where $C_0$ is the nugget value; C is the arch height; $C_0$ + C is the abutment value; and a is the associated length (variation).

### 2.3.2. Drought Index Calculation

Meteorological Drought Index

The SPEI takes into account both precipitation and evapotranspiration elements and has the characteristics of multiple time scales. Compared with the Standardized Precipitation Index (SPI), which only considers rainfall, the SPEI performs better in meteorological drought monitoring [47]. The SPEI was calculated separately for 1–72 month time scales with the following equations:

$$D_i = P_i - PET_i, \tag{4}$$

$$X_i^k = \sum_{i-k+1}^{i} D_i, \tag{5}$$

$$W = \begin{cases} \sqrt{-2\ln(1 - F(x))}, & 1 - F(x) \le 0.5 \\ \sqrt{-2\ln(F(x))}, & 1 - F(x) > 0.5 \end{cases} \tag{6}$$

$$SPEI = \begin{cases} W - \frac{c_0 + c_1 W + c_2 W^2}{1 + d_1 W + d_2 W^2 + d_3 W^3}, & P \le 0.5 \\ \frac{c_0 + c_1 W + c_2 W^2}{1 + d_1 W + d_2 W^2 + d_3 W^3} - W, & P > 0.5 \end{cases}, \tag{7}$$

where $P_i$ is monthly precipitation; $PET_i$ is potential evapotranspiration; $D_i$ is the difference between monthly precipitation and potential evapotranspiration; $X_i^k$ is the value of $D_i$ at different time scales; $k$ is the time scale; $i$ is the number of months; and $F(x)$ is the probability distribution of the variable. In addition, $c_0 = 2.515517$, $c_1 = 0.802853$, $c_2 = 0.010328$, $d_1 = 1.432788$, $d_2 = 0.189269$, and $d_3 = 0.001308$ [12]. Equation (7) follows the polynomial solution of Abramowitz et al. [48].

Groundwater Drought Index

In this study, we refer to the normal quantile transformation (NQT) method applied by Bloomfield, John, and Marchant [24] to calculate the nonparametric standardized groundwater index (SGI). NQT is a method used to transform data to approximately obey a normal distribution [49]. Its main purpose is to make the data more consistent with the assumption of a normal distribution and to minimize the effect of outliers in the data. The normal distribution transformation steps are as follows:

a.    Sort the groundwater sequences for individual grid points from largest to smallest;
b.    Perform a rank (rank) transformation on the sorted data by replacing the original value of each data point with its percentile rank in the sorted data (see Equation (5));
c.    Use a CDF with a standard normal distribution (mean 0, standard deviation 1) to convert the percentile rankings to normally distributed percentile values, and the replaced series is the standardized groundwater drought index [49]. This approach

is employed to transform the percentile rankings into a normally distributed set of percentile values. As such, the resultant index values facilitate the comparison of drought conditions over different time periods and geographical areas, regardless of the original data distribution.

$$R(X) = \frac{L(X) - 0.5}{n}, \tag{8}$$

where $L(X)$ denotes the order of X in the sorted data.

### 2.3.3. Run Theory and Threshold Method

Run theory is commonly used to identify drought events and extract drought characteristic variables [50–53]. It can be applied to different drought indices, but the selection of the drought event threshold needs to be careful [54]. Given the thresholds $R_0$ and $R_1$, when the drought index is less than $R_0$, it is a negative excursion, and when it is greater than $R_0$, it is a positive excursion. If the duration of negative trips exceeds a certain length (usually 2 months or more) [55], a drought event is considered to have occurred and is labeled as a drought; if the negative trip lasts only 1 month and the drought index $< R_1$, drought is also considered to have occurred and is labeled as a drought event; if the interval between two droughts is only 1 month and the drought index $< 0$, the two negative excursions are combined as a drought event; and if none of the above three conditions are satisfied, the event cannot be labeled as a drought event [56]. In this study, the drought frequency (DF) is determined by counting the number of times mild or more severe droughts occurred during the research period. The duration from the start to the end of a drought event is referred to as the drought duration (DD) [57]. During a drought, the cumulative sum of the differences between the drought index values and the drought threshold represents the drought severity (DS). The specific steps for extracting the drought features are as follows:

a. Drought index $< R_0$ is initially recognized as a drought event;

b. Eliminate short-term weak drought events with a drought duration of only 1 month and a drought index $> R_1$;

c. When the interval between the first and last two droughts is only 1 month and the drought index of the interval month is $<0$, then two slightly intermittent droughts are fused into one drought event;

d. Calculate the cumulative frequency F, average duration $\overline{D}$, and average severity $\overline{S}$ of the drought event.

### 2.3.4. Quantification of Drought Propagation

For each pixel, multiple meteorological droughts and groundwater drought events may occur, so the two types of drought events need to be matched first. In this study, if a groundwater drought event ($G_i$) occurs between two consecutive meteorological drought events ($M_i$ and $M_j$) and is closer to $M_i$, then G is matched with Mi, otherwise $G_i$ is considered unmatched or may be caused by non-meteorological factors. After successfully pairing two types of drought events, the peak time (PT), peak intensity (PI), and drought duration (DD) are calculated, respectively. To further understand the propagation process of meteorological drought to groundwater drought, including cumulative [58], delayed [59], attenuation [55], and prolonged [60] aspects, this study calculated the peak lag time (PLT), attenuation scale factor (ASF), and extension scale factor (ESF) between meteorological and groundwater droughts across various meteorological drought time scales (Supporting Information Figure S2). PLT, ASF, and ESF are quantitative indicators of delayed, attenuation, and prolonged effects, respectively. We also provide a summary and detailed explanation of drought terms (Supporting Information Table S2). The formulas for the three quantitative indicators of drought transmission are as follows:

$$PLT = PT_g - PT_m, \tag{9}$$

$$ASF = \frac{PI_g}{PI_m}, \tag{10}$$

$$ESF = \frac{DD_g}{DD_m}. \tag{11}$$

## 3. Results

### 3.1. Spatial and Temporal Variations in Groundwater Depth

#### 3.1.1. Spatial Patterns of Groundwater Depth

Based on the results of the STK method, the annual average of groundwater depth was further calculated to map the spatial distribution of groundwater depth (GWD) in 2005, 2010, 2015, and 2020, respectively (Figure 2a). Spatially, the shallow groundwater depth in the NCP has a zoning pattern of becoming shallower from the western mountainous areas to the central piedmont plains and the eastern alluvial plains. Figure 2b shows the quartiles and outliers of GWD grouped by year in each province. Specifically, GWD in Beijing has increased continuously since 2005, with the depth range expanding from 5~30 m to 5~40 m. Changes in GWD in Hebei Province are more complicated, with a deep groundwater table in the south of the province and a clear trend of increase, whereas the north of the province is relatively stable. In Henan Province, the GWD is mainly concentrated at 0~10 m, but the outlier value can exceed 40 m. The groundwater depth in Shandong Province is relatively stable. In Tianjin Municipality, groundwater depth is mainly in the range of 0~5 m. It is necessary to be vigilant, as GWD has decreased since 2005 and the harm of sea encroachment and land salinization cannot be ignored. Over the past 17 years, the area covered by groundwater depths of 0~5 m in Henan Province has decreased by 70%, while the areas with groundwater depths exceeding 20 m in Beijing and Hebei have expanded eastward. Local groundwater depression cones have increased and shifted from centralized to decentralized features.

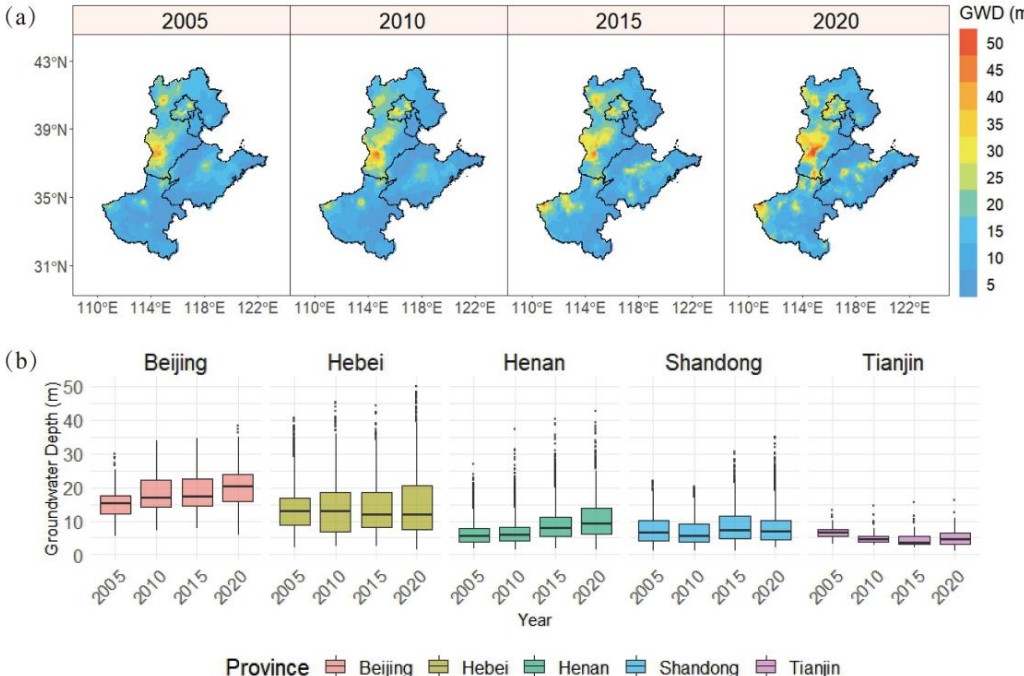

**Figure 2.** (**a**) Spatial distribution of shallow groundwater depth in NCP; (**b**) boxplot of groundwater depth in different provinces grouped by year.

#### 3.1.2. Long-Term Dynamics of Groundwater Depth

Overall, groundwater in the NCP was in a wet stage before 2013. Since January 2013, there has been a persistent increment in groundwater depth across the NCP, indicative of a pronounced decline in the water table (Figure 3). From 2015 to 2021, the regional

groundwater level was at an extremely low level compared to the last 17 years, and the groundwater storage also showed an abnormally low level for the same period. The groundwater "drought" does not abate until the second half of 2021, and water levels rapidly return to wet period levels due to extreme rainfall events.

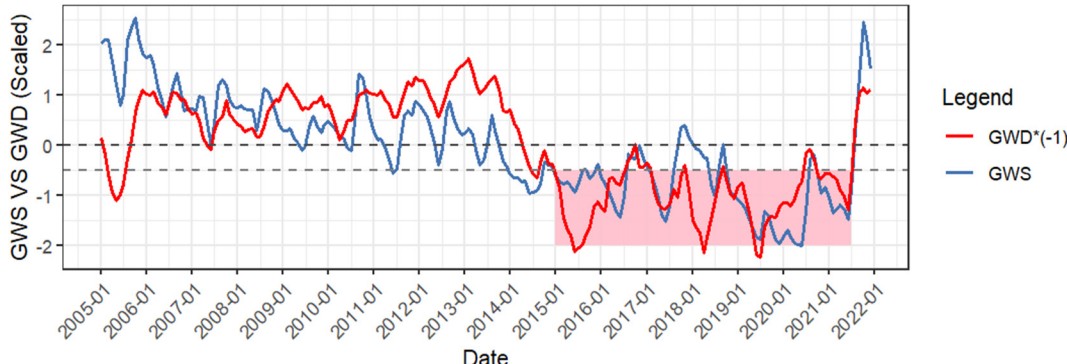

**Figure 3.** Temporal consistency of groundwater depth and storage in NCP. The GWD* is multiplied by −1, so it can be regarded as the relative water level, and both GWD* and GWS are standardized.

Figure 4 shows the multiyear average, coefficient of variation, long-term trend, and significance of groundwater depth in the NCP. The long-term level of groundwater depth has a certain negative correlation with the coefficient of variation (R = −0.36), i.e., the greater the depth, the smaller the coefficient of variation. Deep groundwater levels indicate a thicker vadose zone, and the filtering effect of the vadose zone leads to a lower sensitivity of deep groundwater to climatic variations. During the period from 2005 to 2021, GWD over two-thirds of the regions increased at a rate of 0~0.1 m/a, while GWD in the east coastal region decreased at a rate of −0.05~0 m/a.

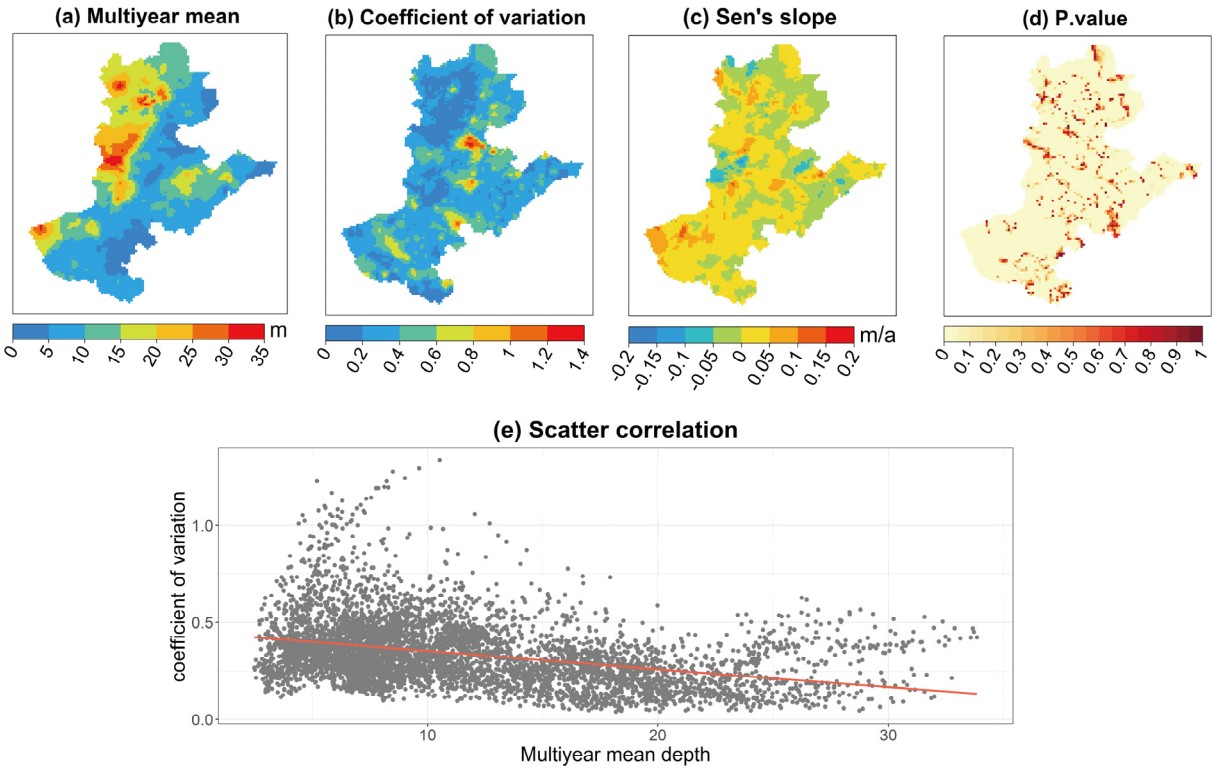

**Figure 4.** Spatial and temporal patterns of groundwater depth in NCP. (**a**) Multiyear mean; (**b**) coefficient of variation; (**c**) long-term trend; (**d**) significance of the trend; (**e**) scatter plot of the correlation between multiyear mean depth and coefficient of variation.

Further analyses were conducted on the seasonal characteristics of groundwater across different land use types, which involved calculating the average groundwater depths and creating monthly box plots (Figure 5). Seasonal dynamics of groundwater in different land surfaces are characterized by "shallow in winter and deep in summer". Significant seasonal variations were observed in forests, cultivated land, water bodies, and unused lands, whereas the seasonal changes in urban and grasslands were less pronounced. Although the NCP experiences the majority of its rainfall from June to September, groundwater levels during this period do not show an immediate corresponding increase, instead remaining at lower levels. This phenomenon can be attributed to two primary factors: Firstly, the recharge process of rainfall to groundwater is affected by canopy interception and slow infiltration of soil, which leads to the delay of groundwater recharge. Secondly, crops in the summer growing season, especially summer corn, which consumes a lot of water, have a large demand for water resources. This not only increases the amount of groundwater extracted by humans but also increases the amount of groundwater excreted by plants through transpiration and soil capillarity. Consequently, owing to the lag in rainfall-induced recharge and intensified discharge activities during the summer, the groundwater depth reaches its maximum in the month of June annually.

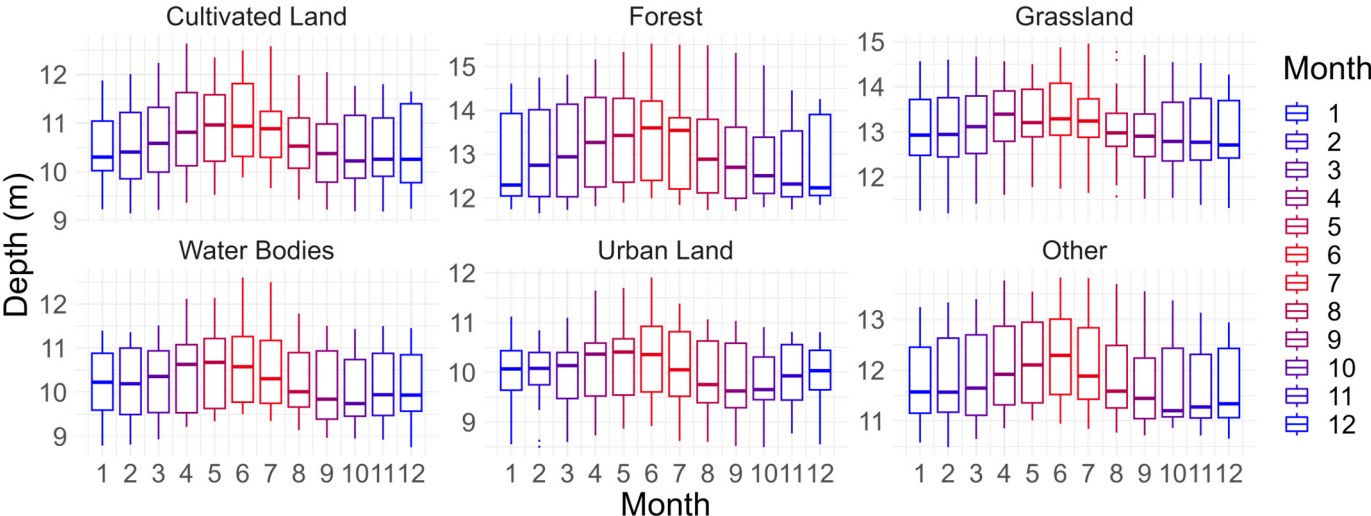

**Figure 5.** Groundwater depth box plots for different subsurface types in NCP.

### 3.2. Groundwater and Meteorological Drought Traits

Figure 6a shows the time series of the SPEI (1–72) with three main influential drought events in North China in the last 17 years, 2006–2007 (Drought I), 2014–2016 (Drought II), and 2019–2021 (Drought III). According to the drought classification table (Supporting Information Table S1), the time proportion of drought occurrence in different degrees was counted. It is found that during the period of 2005–2021 in the NCP, the proportion of drought occurrence time is about 14.71~26.96%. The drought types are mainly light drought and moderate drought, with an average proportion of 15.63% and 3.38%, respectively, and the proportion of time in severe drought is 0.49%. Through the nonparametric method, we further obtained the standardized groundwater index (SGI), which ranges from [−1.89, 1.89]. We found that groundwater was in a drought period from 2014 to 2021. Compared with the regional average SPEI (1–72), it is found that the long time scale SPEI is more common in frequency with the SGI (Figure 6c). From a regional perspective, the correlation between the SPEI and SGI strengthens with the increase in time scale, and the maximum Pearson correlation coefficient can reach 0.43. On the grid scale, the correlation can even reach 0.88 (Supporting Information Figure S1).

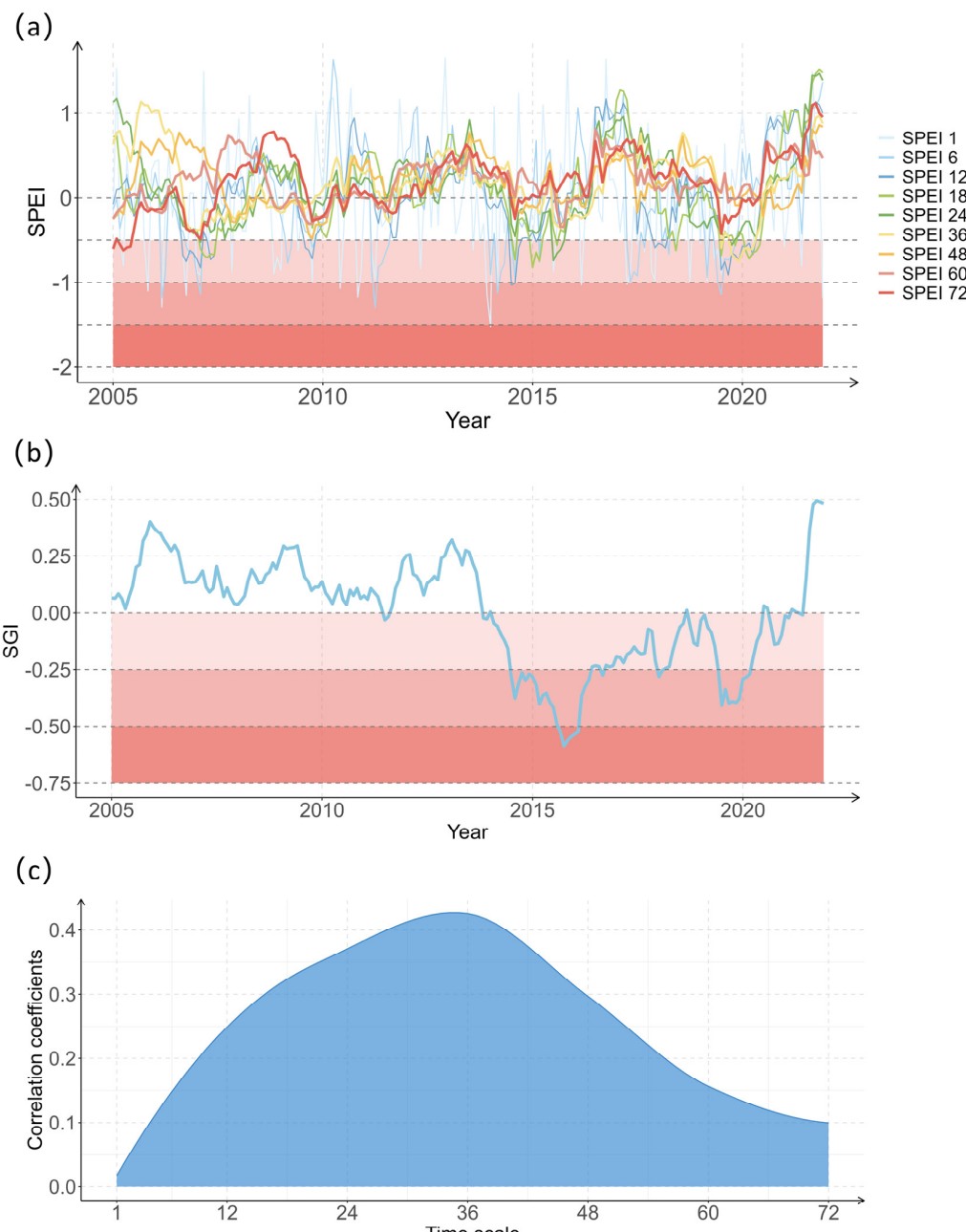

**Figure 6.** (**a**) SPEI (1–72) time series; (**b**) SGI time series; (**c**) SGI-SPEI correlation coefficients vs. time scale.

We have conducted an analysis of meteorological drought characteristics in the NCP at different time scales (Figure 7). Our study reveals significant differences in the frequency of drought events across these scales. On a seasonal scale, droughts are most frequent, with the number of occurrences typically ranging from 15 to 20. At an annual scale, the frequency of droughts decreases, primarily concentrated between 5 and 15. When examining periods exceeding one year, the frequency of droughts is usually less than 5. In terms of duration, as the temporal scale extends, there is an observed increase in the average length of drought events, with the range varying between 0 and 78 months. Longer time scales indicate the influence of more persistent climatic patterns and hydrological conditions. Moreover, the average drought severity spans between 0 and 80, exhibiting an increasing trend with the expansion of temporal scales. It is noteworthy that although the drought severity remains below 30 in over two-thirds of the regions, the eastern NCP notably suffers from higher intensities than the west.

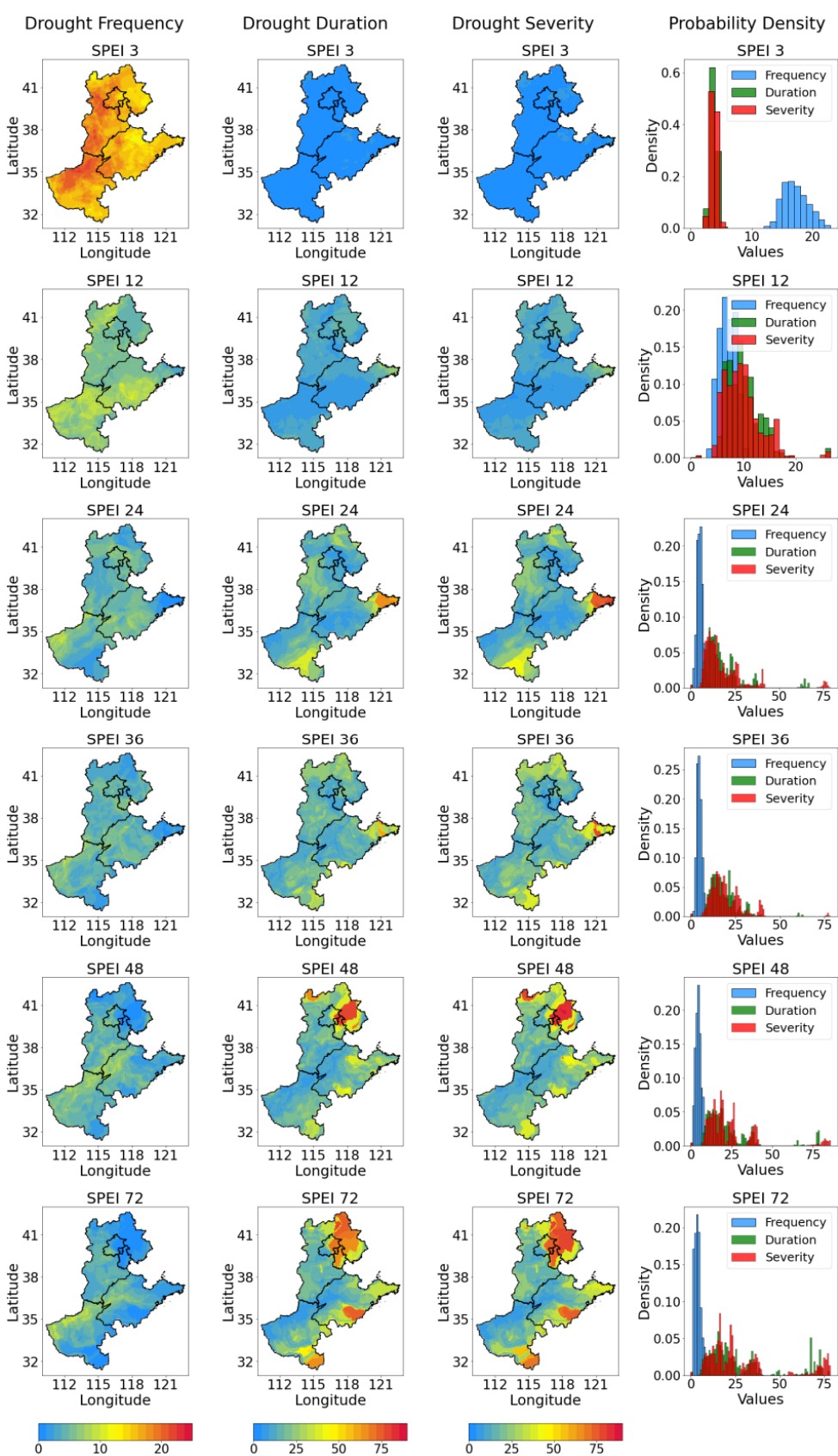

**Figure 7.** Spatial distribution of meteorological drought characteristics. From left to right, the columns represent drought frequency, drought duration, drought severity, and probability density.

According to the groundwater drought characteristics extracted from the SGI as follows (Figure 8), the frequency of groundwater droughts in North China in the past 17 years ranged from [0, 14], the average drought duration from [0, 62], and the average drought severity from [0, 73]. According to the probability density distribution curve, the groundwater drought frequency in the NCP is mainly concentrated at 3~6 times, the average drought duration is concentrated at 10~30 months, and the average drought severity is concentrated at 10~35. Among them, groundwater droughts in Shandong, northeast of Hebei (Chengde, Qinhuangdao), and southwest of Henan (Xinyang, Zhumadian, Zhoukou, Zhengzhou) are characterized by a high number of droughts, a short average duration of a single drought, and low severity, while Beijing, Tianjin, and most of the cities in the western part are characterized by a low number of droughts, a long average duration of a single drought, and high severity.

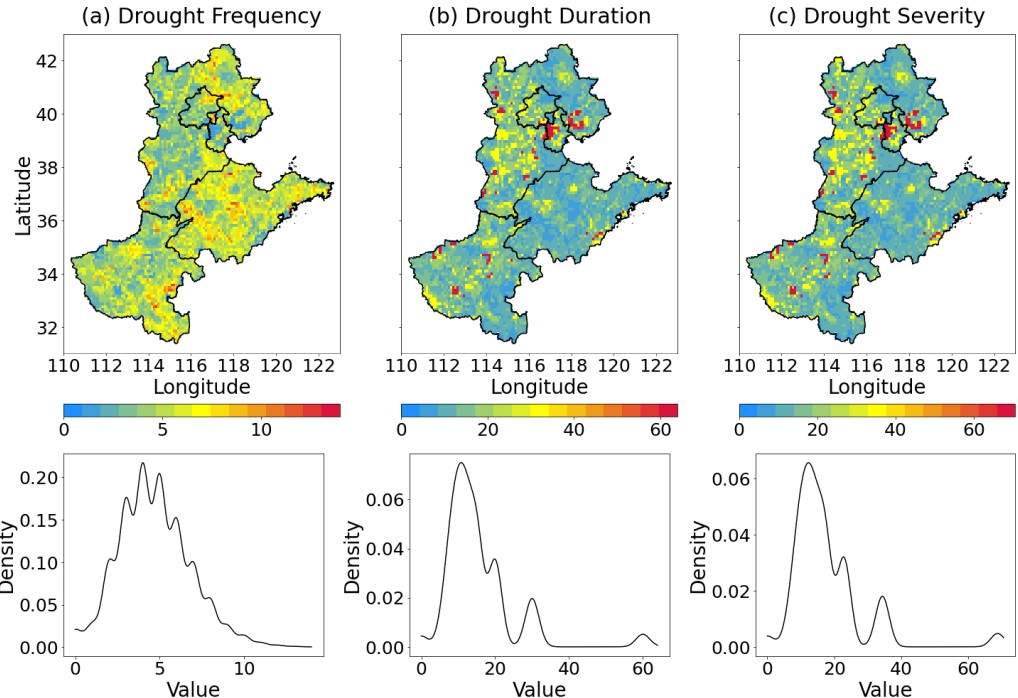

**Figure 8.** Spatial distribution of groundwater drought characteristics. (**a**) Drought frequency; (**b**) drought duration; (**c**) drought severity.

### 3.3. Meteorological–Groundwater Drought Propagation Patterns

Meteorological drought impacts an array of groundwater display characteristics, including cumulative, delayed, attenuation, and prolonged effects (Figure 9). This study reveals how groundwater systems respond to meteorological drought on different time scales. Delayed effects are evident as the influence of meteorological drought on groundwater does not occur instantaneously but rather incurs a time lag ranging from 0 to 60 months. With an extension in SPEI timescales (i.e., cumulative effect increases), the observed lag in groundwater response to meteorological drought intensifies. It should be emphasized that in the northern region of the NCP, the groundwater system's delayed response to surface drought conditions is considerably more pronounced than in the southern region. The attenuation ratio factor provides a quantitative assessment of the decrease in drought severity as it transitions from affecting atmospheric conditions to impacting groundwater. This factor fluctuates between 0 and 3, with values under 1 (highlighted in red areas) denoting significant attenuation. The strongest attenuation of meteorological drought impact on groundwater is observed at the 24-month SPEI scale. This could be attributed to the inertial response of groundwater systems to climatic fluctuations, which becomes more apparent over longer temporal scales. The prolonged ratio factor spans from 0 to 16, with values

above 1 (also marked in red areas) indicating that the effects of meteorological drought persist even after the drought event has ended. With the escalation of the SPEI temporal scale, the prolonged effect diminishes, with a more pronounced persistence of these effects in the northern regions compared to the southern.

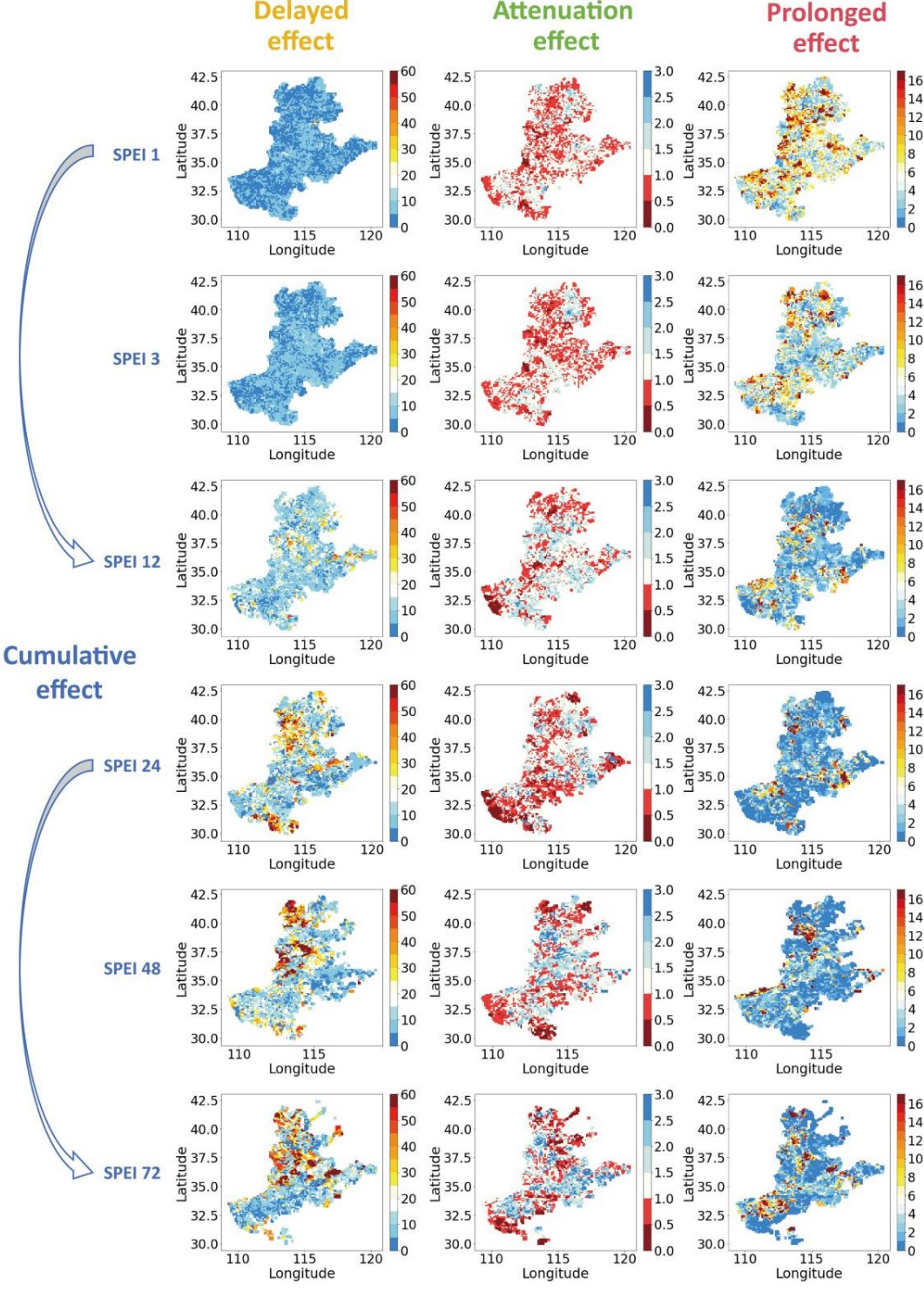

**Figure 9.** The cumulative effect, delayed effect, attenuation effect, and prolonged effect of the propagation of meteorological drought to groundwater drought.

## 4. Discussion

Similar to Entekhabi [55], our study is based on long-term observational data. Meanwhile, we focus on how meteorological drought affects groundwater systems. On one hand, we expanded the actual measurement site data to continuous grids, investigating the basic situation of groundwater in the NCP for nearly 17 years; on the other hand, we quantified the characteristics of single-type drought and the propagation characteristics between the two types of drought.

Drought indices based on the SPEI and SGI successfully captured meteorological drought events in the North China Plain. The three major drought events in the past 17 years occurred in 2006–2007, 2014–2016, and 2019–2021, which is consistent with the "China Flood and Drought Disaster Bulletin" (http://www.mwr.gov.cn/sj/tjgb/zgshzhgb/) and the meteorological disaster database constructed by Hu Duanmu et al. [61] using natural language processing technology. Before 2014, relatively short-term meteorological drought events did not trigger groundwater drought, but the meteorological drought from 2014 to 2016 caused severe groundwater drought, which was not recovered in the following five years. This indicates that the groundwater system has a strong drought resistance, but once it exceeds its capacity, it is difficult to recover from drought [62].

The spatial consistency between meteorological drought characteristics and groundwater drought characteristics is not high, but the temporal correlation is significant. In the process of drought propagation from meteorological to groundwater drought, the frequency of drought occurrences decreases, while the duration and severity of drought increase, which is similar to the results of Han, Zhiming et al. [63], Gong et al. [64], Yin et al. [65], Zhao et al. [17], etc. The process of drought propagation is complicated, and the spatial distribution of drought propagation characteristics is caused by multiple factors such as climatic conditions, surface physical properties [66], vadose zone thickness and structure (Supporting Information Figure S2), and human activities. Compared with the southern part of the plain, the northern part of the plain has less precipitation and lower temperatures, and the irrigated agriculture depends more on groundwater. Studies have shown that crops in the northern part of the NCP are more vulnerable to drought stress than those in the southern part of the plain, which further forces humans to mitigate crop loss through continuous pumping irrigation. As a result, groundwater drought is more severe and has a longer delay time. In addition, Wang, Fei, et al. [27] demonstrated the impact of teleconnection factors on groundwater drought in the North China Plain.

There are already global assessments of drought propagation [67], indicating that hydrological systems respond to meteorological drought faster than vegetation systems. However, these studies did not involve the North China Plain nor did they deeply analyze the response of groundwater systems to meteorological drought. In drought propagation research, previous studies mostly relied on methods like the maximum correlation coefficient method [64] and traditional lag cross-correlation analysis [55], which depend on the linear relationship between variables. If the propagation of drought involves nonlinear processes, these methods have limitations in quantifying the drought transmission time. Additionally, the drought delayed time calculated by Entekhabi [55] is based on the difference in the start time of groundwater drought and meteorological drought, and the degree of drought attenuation is calculated based on the area enclosed by the drought process curve and the drought threshold. However, the start time of drought is not always so clear, and the selection of different drought indices and different threshold levels will bring uncertainty to the results. Therefore, our study avoids this problem by drought matching and peak extraction methods. The propagation from meteorological drought to groundwater drought may also involve agricultural drought and hydrological drought, and most studies show that the drought delayed time between surface systems is on a monthly or seasonal scale [60,68,69], while the delayed time from meteorological drought to groundwater drought usually exceeds one year, and in this study, it can be up to 60 months.

Due to the lack of real pumping data and collinearity between variables [70], a limitation of our study is that groundwater extraction activities are not considered. The impact of

large cities and intensive agricultural activities on groundwater systems cannot be ignored. However, quantifying actual agricultural water use and other anthropogenic impacts is a complex challenge, although indirect methods have been used to assess the impact of human activities on groundwater systems [64]. We look forward to quantifying the role of human activities based on more accurate groundwater extraction data in our future studies. Furthermore, during the STK interpolation process, we opted for the commonly used spherical model without further comparing the influence of alternative models such as the exponential or Gaussian models. While the chosen model adequately fitted the interpolation, a comprehensive analysis involving various models could provide a more nuanced understanding of the results. Finally, this paper only explores the relationship between meteorological drought and groundwater drought. In fact, the transmission from meteorological drought to groundwater drought also involves agricultural drought, hydrological drought, and socio-economic drought. The relationship between various types of drought is complicated, and groundwater drought may also further affect socio-economic and agricultural drought.

## 5. Conclusions

In this study, the spatial and temporal patterns of groundwater and meteorological drought response in the NCP were deeply investigated. The missing and discontinuous problems in the monitoring data were dealt with by the spatiotemporal kriging interpolation method, and more complete and reliable monthly raster data of groundwater depth were obtained, which provided a basis for analyzing the spatiotemporal evolution of groundwater.

The results show that the depth of shallow groundwater in the NCP has an obvious spatial zoning pattern, gradually becoming shallower from the western mountainous areas to the eastern alluvial plains. In time, most of the groundwater depths in the region showed an increasing trend between 2005 and 2021. Especially in Henan Province, the area covered by a 0–5 m depth of groundwater has decreased significantly, while the area covered by a more than 20 m depth of groundwater in Beijing and Hebei has been expanding. In terms of seasonal changes, the characteristic of "shallow in winter and deep in summer" is common in different land use types.

In terms of drought characteristics, two indicators, SPEI and SGI, were analyzed. It was found that the North China Plain has experienced three important meteorological drought events in the past 17 years. The standardized groundwater drought index (SGI) showed that the region was generally in a drought state during 2014–2021. Correlation analysis showed that the effect of meteorological drought on groundwater drought strengthens with an increasing time scale. Additionally, our study reveals that the propagation of meteorological drought to groundwater drought exhibits cumulative, delayed, attenuation, and prolonged effects with spatiotemporal heterogeneity. The delayed effect intensifies with increased temporal scales, particularly in the NCP's northern regions, while the attenuation effect remains relatively stable, and the prolonged effect decreases. It is noteworthy that a significant attenuation effect is observed in more than half of the study areas, and the prolonged effect is primarily associated with intra-annual meteorological drought events.

Compared to previous research [17,25,27,28,71], this study systematically investigates the response patterns of groundwater to meteorological drought, specifically quantitatively analyzing the cumulative, delayed, attenuation, and prolonged effects in the drought propagation process. The results provide valuable scientific evidence for understanding the complex relationship between groundwater and meteorological drought, offering significant guidance for regional water resource management and the formulation of drought mitigation strategies.

**Supplementary Materials:** The following supporting information can be downloaded at: https://www.mdpi.com/article/10.3390/w16030501/s1, Table S1. Meteorological drought classification based on SPEI; Table S2. Interpretation of drought terms; Table S3. Summary of drought characteristic variables and propagation characteristic indexes; Figure S1. The maximum Pearson correlation coefficient between SPEI and SGI at different time scales at the grid scale; Figure S2. Schematic diagram of the drought propagation process; Figure S3. The relationship between groundwater drought characteristics and groundwater depth. From left to right: (a) drought frequency—groundwater depth; (b) drought duration—groundwater depth; (c) drought severity—groundwater depth. Long-term conditions of groundwater depth in the unconfined aquifer are equivalent to the thickness of the vadose zone. References [72–83] are cited in the supplementary materials.

**Author Contributions:** Y.Z. formulated the problem and designed the experiment; Y.C. analyzed the data and wrote the paper; J.T., Z.T., L.W. and X.Y. contributed to the validation work and language editing. All authors have read and agreed to the published version of the manuscript.

**Funding:** This research was funded by the National Key R&D Program of China (Grant No. 2022YFC3002804), the National Natural Science Foundation of China (Grant No. 42071327 and Grant No. 42330506), and the CAS International Partnership Program (Grant No. 183311KYSB20200015).

**Data Availability Statement:** All data used in this study are available from the following websites: groundwater in situ monitoring data (http://www.bom.gov.au/water/groundwater/); groundwater storage (https://giovanni.gsfc.nasa.gov/giovanni/#service=TmAvMp&starttime=&endtime=&dataKeyword=Groundwater%20storage); climate data (http://data.tpdc.ac.cn/); and land cover data (https://www.resdc.cn/DOI/doi.aspx?DOIid=54).

**Conflicts of Interest:** The authors declare no conflicts of interest.

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
