# Peer review of "Understanding the Propagation of Meteorological Drought to Groundwater Drought: A Case Study of the North China Plain"

_water, doi:10.3390/w16030501_

Round 1
Reviewer 1 Report
Comments and Suggestions for Authors
The idea of the research is good, but there is missing information that would clarify the results or indicate that more work is needed. First, to adequately model the plain, you have to consider surface and groundwater from the surrounding mountains. It is possible that glacial and/or snowmelt will contribute for a period of time to recharging groundwater, but that once glaciers are gone as a source of discharge, their impact will be lost. The geography and snow levels of the surrounding mountains and their impact must be explained, especially as rising global temperatures are shrinking glaciers. Without any description of the surrounding mountains, their water supplies, and how these feed groundwater, the groundwater on the plain cannot be fully understood.
Second, groundwater removal from major cities and agricultural lands must be included. There are several significant cities in the study area and their removal of groundwater was not explained. THis is exacerbated by meteorological drought, so it must be accounted for.
Define in the introduction what meteorological drought is as compared to the other droughts. Also, make sure all acronyms are defined and all units are defined.
You state that your study uses data based on observations, which is partially true. You also use models and interpolation methods to fill in missing data. With no actual data to compare modeled data to, this is still somewhat modeling. Be clear in that assertion.
In the abstract and discussion, you give numbers for attenuation effect and SGI without explaining what they mean in the context of drought. Perhaps a table would provide information to make the numbers meaningful.
In equations 5-7, explain how co-c2 and d1-d3 were determined.
It would be helpful to make the figures bigger, especially the many small ones with color ranges.
Comments on the Quality of English Language
Good with some minor editing.
Author Response
Dear Reviewers:
Thank you for your letter and for the comments concerning our manuscript entitled “Understanding the propagation of meteorological drought to groundwater drought: a case study of the North China Plain” (ID: water-2841883). They were all valuable and very helpful for revising and improving our paper. We have carefully made correction. All revisions to the manuscript were marked up using the “Track Changes” function using MS Word. The main corrections in the paper and the responds to the reviewer’s comments are as following:
Point 1: The idea of the research is good, but there is missing information that would clarify the results or indicate that more work is needed. First, to adequately model the plain, you have to consider surface and groundwater from the surrounding mountains. It is possible that glacial and/or snowmelt will contribute for a period of time to recharging groundwater, but that once glaciers are gone as a source of discharge, their impact will be lost. The geography and snow levels of the surrounding mountains and their impact must be explained, especially as rising global temperatures are shrinking glaciers. Without any description of the surrounding mountains, their water supplies, and how these feed groundwater, the groundwater on the plain cannot be fully understood.
Response 1: Thanks to reviewer for this suggestion. The North China Plain is located in the warm temperate semi-arid monsoon climate area, and the summer precipitation recharge is the most important source of shallow groundwater recharge in the North China Plain. In addition, there is also a small amount of river infiltration recharge and mountain lateral runoff recharge, and the contribution of glaciers in this study area is almost negligible. This crucial detail was not sufficiently emphasized in the original manuscript. Therefore, we have added this information to the section describing the study area to clarify the unique hydrological context of the North China Plain and to better explain the mechanisms of groundwater recharge in this region. (Line 143-146)
Point 2: Second, groundwater removal from major cities and agricultural lands must be included. There are several significant cities in the study area and their removal of groundwater was not explained. This is exacerbated by meteorological drought, so it must be accounted for.
Response 2: Thank you very much for your constructive suggestions. We acknowledge this lack of consideration and understand the importance of this aspect in our research. It is critical to consider the impact of human factors such as urban and agricultural extraction of groundwater on the water table, especially in the context of the North China Plain. The impact of large cities and intensive agricultural activities on groundwater systems, coupled with the effects of meteorological drought, cannot be ignored. However, modeling actual agricultural water use and other anthropogenic impacts is a complex challenge. Due to the difficulty in obtaining real-world pumping data and the complexity of climate-human interactions, we were unable to include human activity in our study. Therefore, we have added to the discussion section of our manuscript a comprehensive discussion of the potential uncertainties and the effects associated with these factors. This study focuses on the propagation of meteorological drought to groundwater drought. In future studies, we will further quantify the role of human activities in drought transmission.
Point 3: Define in the introduction what meteorological drought is as compared to the other droughts. Also, make sure all acronyms are defined and all units are defined.
Response 3: Thank you for pointing out the necessity for clarity in our manuscript. We have revised the introduction to include a precise definition of meteorological drought [1], distinguishing it from hydrological, agricultural, and socioeconomic droughts. (Line 35-40)
To ensure complete clarity, all acronyms used in the manuscript have now been defined upon their first occurrence. Additionally, we have meticulously reviewed the document to verify that all units of measurement are consistently defined and used.
Point 4: You state that your study uses data based on observations, which is partially true. You also use models and interpolation methods to fill in missing data. With no actual data to compare modeled data to, this is still somewhat modeling. Be clear in that assertion.
Response 4: Thank you for the valuable feedback. We appreciate the reviewer's attention to the nature of our data sources. We acknowledge that our study involves a combination of observed data and modeling techniques, including interpolation methods to address missing data. We have revised the description of the data. (Line 456)
Point 5: In the abstract and discussion, you give numbers for attenuation effect and SGI without explaining what they mean in the context of drought. Perhaps a table would provide information to make the numbers meaningful.
Response 5: Thank you for your constructive feedback. We appreciate your suggestions in the summary and discussion to improve the clarity of the article. We understand the importance of ensuring that readers can fully understand the significance of these numbers and variables in the context of the drought. Therefore, we have added a table in the supplementary material to make it easier for readers to understand the meaning of the numbers and variables in the context of the drought (Table S2).
Point 6: Figure 1,5, and 6 must be larger so that the reader can better observe the image
Response 6: Thank you for your suggestion. We have resized Figures 1, 5, and 6 to be larger, improving the visibility for readers to better observe the images. This adjustment is aimed at enhancing the overall readability and clarity of the figures. We have made the elements in the figure clearer, the image has been adjusted larger, and the resolution has been increased to 800dpi (previously 600). We appreciate your valuable feedback.
Point 7: In equations 5-7, explain how co-c2 and d1-d3 were determined.
Response 7: Thank you for your careful review of this article. Firstly, we would like to explain that equations 4-7 are referenced from the work of Vicente-Serrano et al. [2], who are the developers of the SPEI index. Formula 7 is designed to normalize the probability distribution function F(x). While the cumulative distribution function of a normal distribution can be solved using table lookup, this method is inefficient. Another approach is based on the polynomial computation method proposed by Abramowitz et al. [3] in the book "Handbook of Mathematical Functions with Formulas, Graphs, and Mathematical Tables" (Page 933, https://personal.math.ubc.ca/~cbm/aands/frameindex.htm). This method ensures a computational absolute accuracy of up to 4.5×10-4, with five fixed constant parameters available for easily application. Thanks again for your suggestions, we have added more equation descriptions (Line 239-240).
Point 8: It would be helpful to make the figures bigger, especially the many small ones with color ranges.
Response 8: We sincerely appreciate your valuable feedback. Based on your insightful suggestions, we have made enhancements to the size, font size, and resolution of the graphics, specifically in Figure 1, 4, 5, and 6. These improvements result in more visually appealing and clearer graphics, addressing the concern you raised.
Special thanks to you for your good comments.
There is no doubt that your responsible detailed suggestions and high-quality reviews have significantly improved the quality of the article. We appreciate for your warm work earnestly, and hope that the correction will meet with approval.
Once again, thank you very much for your comments and suggestions.
Reference
- 1. "Encyclopedia of climate and weather (2nd edition)." Reference Reviews 26 (2012): 36-37. 10.1108/09504121211233844. https://doi.org/10.1108/09504121211233844.
- 2. Vicente-Serrano, S. M.; S. Beguería and J. I. López-Moreno. "A multiscalar drought index sensitive to global warming: The standardized precipitation evapotranspiration index." Journal of Climate 23 (2010): 1696-718. https://doi.org/10.1175/2009JCLI2909.1.
- 3. Abramowitz, M.; I. A. Stegun and D. Miller. "Handbook of mathematical functions with formulas, graphs and mathematical tables (national bureau of standards applied mathematics series no. 55)." Journal of Applied Mechanics 32 (1965): 239-39.

Reviewer 2 Report
Comments and Suggestions for Authors
1.The introduction needs to explicitly state the existing research gap that this study aims to address in the context of groundwater and meteorological drought interactions. Clearly articulate what is not well-understood or explored in the current literature.
2.The section discussing drought indicators is extensive and somewhat convoluted. Consider condensing the information, focusing on the most relevant indicators for meteorological and groundwater drought linkages, and briefly explaining their applicability.
3.When introducing various groundwater drought indices, provide a brief rationale for their development and explain how each contributes to understanding the propagation and severity of groundwater drought. This will help readers grasp the significance of these indices.
4.The study mentions statistical theories, Spatio-Temporal Kriging interpolation, and the Mann-Kendall trend test without detailing their application. Expand on how these methods were employed, their strengths, and potential limitations. This will enhance the study's transparency and reproducibility.
5.Acknowledge and discuss the limitations and assumptions of the study, such as the constraints of the chosen methodologies or potential biases in the data. This transparency will strengthen the credibility of your findings.
6.Ensure consistency in the use of terms such as "delayed effects," "prolonged effects," and "attenuation effects." Provide precise definitions and illustrate their significance in the context of your study to avoid ambiguity.
7. The study describe differences between northern and southern regions but lacks depth. Provide more insights into these variations, offering explanations or hypotheses for the observed distinctions in groundwater drought effects
Author Response
Dear Reviewers:
Thank you for your letter and for the comments concerning our manuscript entitled “Understanding the propagation of meteorological drought to groundwater drought: a case study of the North China Plain” (ID: water-2841883). They were all valuable and very helpful for revising and improving our paper. We have carefully made correction. All revisions to the manuscript were marked up using the “Track Changes” function using MS Word. The main corrections in the paper and the responds to the reviewer’s comments are as following:
Point 1: The introduction needs to explicitly state the existing research gap that this study aims to address in the context of groundwater and meteorological drought interactions. Clearly articulate what is not well-understood or explored in the current literature. 

Response 1: Thanks to reviewer for this suggestion. In response to your suggestion, we have revised the introduction to explicitly state the existing research gap that this study aims to address in the context of groundwater and meteorological drought interactions. We have articulated the aspects that are not well-understood or explored in the current literature (Line 102-109). We believe these revisions enhance the clarity and significance of our research, and we hope they meet your expectations. If you have any further recommendations or specific points you would like us to address, please feel free to let us know. Thank you again for your thoughtful review.
Point 2: The section discussing drought indicators is extensive and somewhat convoluted. Consider condensing the information, focusing on the most relevant indicators for meteorological and groundwater drought linkages, and briefly explaining their applicability.
Response 2: Thanks for the reviewer’s detailed suggestions. We admit that this part is not concise enough. In response, we highlight the drought index most closely associated with this study and briefly summarize its applicability (Line 55-61). In addition, the supplementary material (Table S3) provides a table that provides an overview of the meteorological and groundwater drought indicators currently used in drought studies, as well as a summary of their rationale and advantages and disadvantages. Our goal is to improve clarity and simplicity while providing a focused explanation of its applicability. We appreciate your guidance and will ensure a more streamlined presentation in the revised manuscript.
Point 3: When introducing various groundwater drought indices, provide a brief rationale for their development and explain how each contributes to understanding the propagation and severity of groundwater drought. This will help readers grasp the significance of these indices.
Response 3: Thank you for your constructive feedback. We acknowledge the importance of providing a brief rationale for the development of groundwater drought indices and explaining their contributions to understanding propagation and severity. In response to your suggestion, we have enhanced the introduction section by succinctly outlining the development motivations behind each groundwater drought index and highlighting their specific roles in elucidating the dynamics and severity of drought propagation (Line 62-98).
Point 4: The study mentions statistical theories, Spatio-Temporal Kriging interpolation, and the Mann-Kendall trend test without detailing their application. Expand on how these methods were employed, their strengths, and potential limitations. This will enhance the study's transparency and reproducibility.
Response 4: Thank you for your insightful advice. We have included the applications, advantages, and limitations of each method in the Methods section (Line 202-211, Line 230-233, Line 254-258, Line 262-264). The transparency and reproducibility of our research has further increased.
Point 5: Acknowledge and discuss the limitations and assumptions of the study, such as the constraints of the chosen methodologies or potential biases in the data. This transparency will strengthen the credibility of your findings.
Response 5: We sincerely appreciate your insightful suggestions. We provide a comprehensive discussion of the limitations of the chosen method and potential biases in the data in the revised manuscript (Line 511-526). By addressing these limitations openly, we aim to provide a transparent view of research boundaries and potential sources of uncertainty. This addition not only enhances the credibility of our findings, but also contributes to a more robust interpretation of the results. We sincerely thank you for your constructive guidance throughout the review process, and we believe that these revisions have contributed positively to the overall quality of our manuscript.
Point 6: Ensure consistency in the use of terms such as "delayed effects," "prolonged effects," and "attenuation effects." Provide precise definitions and illustrate their significance in the context of your study to avoid ambiguity.
Response 6: Thanks to reviewer for this suggestion. To enhance clarity, we have revised the manuscript to ensure consistency in the use of terms such as "delayed effects," "prolonged effects," and "attenuation effects" (Line 22, Line 119-121, Line 303-306). We have also provided precise definitions (Supporting Information Table S2) and offer illustrations to underscore their significance in the context of our study (Supporting Information Figure S2), aiming to eliminate any potential ambiguity.
Point 7: The study describe differences between northern and southern regions but lacks depth. Provide more insights into these variations, offering explanations or hypotheses for the observed distinctions in groundwater drought effects
Response 7: Thank you for your careful review of this article. In the revised manuscript, we have provided additional explanations to better elucidate the observed distinctions in groundwater drought effects between these two regions (Line 478-487). We believe that this enhancement will contribute to a more comprehensive understanding of the study findings.
Special thanks to you for your good comments.
There is no doubt that your responsible detailed suggestions and high-quality reviews have significantly improved the quality of the article. We appreciate for your warm work earnestly, and hope that the correction will meet with approval.
Once again, thank you very much for your comments and suggestions.

Reviewer 3 Report
Comments and Suggestions for Authors
The present work is concerned with the mechanism of meteorological drought propagation to groundwater drought in North China Plain utilizing statistical theories, Spatio-Temporal Kriging interpolation, and the Mann-Kendall trend test during the groundwater period 2005 to 2021. The characteristics and propagation process of drought are further quantified based on drought theory. The spatial and temporal patterns of groundwater and meteorological drought response, and the missing and discontinuous problems in the monitoring data are the main targets. It is determined that shallow groundwater depths follow a zonal pattern from the western mountains to the eastern plains and coastal areas. It is also identified that groundwater drought frequency typically ranges from 3 to 6 times, with an average duration of 10 to 30 months. It is mainly concluded that both delayed and prolonged effects in the northern part are more pronounced than in the southern of the region. Effective drought monitoring and water management are thought to be contributed through the present additions.
The work seems to be publishable in MDPI: Water if a careful attention is given to the following coments;
A)- Abstract should incorporate a sentence pointing to the most novel contribution in terms of future significance.
B)- Language is fine overall, take care of typos and some grammatical deficiencies.
C)- The literature survey is fully linked to the topic, clearly pinpointing to the significance of the study.
D)- In groundwater drought studies, generally the groundwater drought indices based on GRACE and GLDAS (Global Land Data Assimilation System) are used in the literature. In the Abstract and Discussion sections, clarify how this is used in the current analysis.
E) – Some formulas should be referenced properly, if they are taken from the literature.
F)- The North China Plain Features given in Section 2 should be summarized in a Table.
G) - The probability distribution of the variable and parameters in Ref.[11] are used. Are these suitable/optimal for the current analysis?
H) - Some data regression is used for continuous long-series monitoring. Refer to other approximate methods also, such as DOI: 10.32604/cmes.2021.012595.
K) - Why is the standardized groundwater drought index given by Eq.(8)?
L) - Drought Duration and Drought Severity Density values are almost the same. Explain why.
M) - The current work concentrated only on a certain domain of China. Discuss how it can be adjusted to the rest of the world. How can the results be used in the rest of the world?
Comments on the Quality of English LanguageGood
Author Response
Dear Reviewers:
Thank you for your letter and for the comments concerning our manuscript entitled “Understanding the propagation of meteorological drought to groundwater drought: a case study of the North China Plain” (ID: water-2841883). They were all valuable and very helpful for revising and improving our paper. We have carefully made correction. All revisions to the manuscript were marked up using the “Track Changes” function using MS Word. The main corrections in the paper and the responds to the reviewer’s comments are as following:
Point 1: Abstract should incorporate a sentence pointing to the most novel contribution in terms of future significance. 

Response 1: Thank you for the reviewer's suggestion. We have made revisions to lines 24-27 of the abstract to emphasize the most novel contribution of our study. Previous research has rarely systematically quantified the cumulative effect, delayed effect, attenuation effect, and prolonged effect that occur during the drought transmission process. What sets our study apart is our unique approach to understanding the interaction between groundwater and meteorological drought by delving into these four phenomena. This novel contribution holds significant future significance for further research.
Point 2: Language is fine overall, take care of typos and some grammatical deficiencies.
Response 2: Thank you for your careful review of this article. We have made corrections to potential grammar and spelling errors in the manuscript as follows:
Line 20: Removed the extra space;
Line 137: 'lakeland' -> 'lacustrine';
Line 142: Corrected the unit;
Line 152: 'In situ' -> 'In-situ';
Line 162: 'GLDAS2.2' -> 'GLDAS-2.2';
Line 173-175: Corrected punctuation;
Line 235-2383: Fixed punctuation and space errors;
Line 301: Removed the extra space;
Line 334: 'Of' -> 'of';
Lines 358: 'under' -> 'across';
Line 492: 'does' -> 'did'
Point 3: The literature survey is fully linked to the topic, clearly pinpointing to the significance of the study.
Response 3: Thank you for acknowledging the alignment of our literature survey with the study topic. We aimed to provide a comprehensive review that clearly emphasizes the significance of our study within the existing body of knowledge. We appreciate your positive feedback and are pleased that our efforts to establish a strong connection between the literature and the study's importance have been successful.
Point 4: In groundwater drought studies, generally the groundwater drought indices based on GRACE and GLDAS (Global Land Data Assimilation System) are used in the literature. In the Abstract and Discussion sections, clarify how this is used in the current analysis.
Response 4: Thanks for the reviewer’s detailed suggestions. In the Abstract section, we have provided further clarification regarding the role of GLDAS in our study. We have demonstrated that groundwater storage derived from remote sensing and hydrological modeling exhibits consistent dynamic characteristics at the regional scale when compared to groundwater depth data obtained from monitoring wells. The mutual validation between remote sensing and ground-based measurements reinforces the reliability of our research findings. In the Discussion section, we have emphasized that while GWD and GWS demonstrate consistency at a larger scale (typically exceeding 200,000 km2), finer-resolution groundwater drought monitoring still relies on monitoring wells.
Point 5: Some formulas should be referenced properly, if they are taken from the literature.
Response 5: Thank you for helping us to modify it carefully. We have now added the appropriate references to the formula.
Equation 1 Add references: [1]
Equation 2 Add references: [2]
Equation 3 Add references: [3]
Equation 8 Add references: [4]
Point 6: The North China Plain Features given in Section 2 should be summarized in a Table.
Response 6: Thank you for the reviewer's constructive suggestions. In Section 2.1 of our manuscript, we have incorporated a new table providing an overview of the North China Plain region (Line 149-150). This table includes information on the provinces and cities covered in our study, the total area, latitude, longitude, elevation, rainfall, temperature, land cover percentage, number of monitoring wells, major crop types, crop area affected by drought, population facing difficulties due to drought, and the number of large livestock facing water scarcity. We believe that this addition enhances the comprehensiveness of our study and provides valuable context for readers to better understand the North China Plain region.
Table 1 Basic information of the study area
Features |
Description |
Provinces and cities |
Beijing, Hebei, Tianjin, Henan, Shandong |
Total covered area |
538,000 km2 |
Longitude |
110°~123° E |
Latitude |
32°~43° N |
Mean Elevation |
312 m |
Annual mean precipitation |
340~910 mm |
Annual mean temperature |
10~15 ℃ |
Proportion of land cover |
Cropland (85.2%), Forest (9.8%), Grassland (2.2%), Urban areas (2.1%), Water bodies (0.6%), Bare areas(0.1%) |
Monitoring well count |
1826 |
Staple crop |
winter wheat, summer corn, spring corn |
Area of crops most affected by drought |
4.3 million hectares (2009) |
Population with drinking water difficulties due to drought |
3.3 million (2010) |
Large livestock with difficulty in drinking water due to drought |
1.3 million (2006) |
Point 7: The probability distribution of the variable and parameters in Ref.[11] are used. Are these suitable/optimal for the current analysis?
Response 7: Thank you for your careful review of this article. In drought research, there exists a range of indices that can characterize meteorological drought, including but not limited to the Standardized Precipitation Index (SPI), Standardized Precipitation Evapotranspiration Index (SPEI), Palmer Drought Severity Index (PDSI), Percentage of Precipitation Anomaly (PAP), China-Z Index (CZI), and the Supply-Demand Water Index (SWAP). Each of these indices has its strengths and limitations. Specifically, the SPEI is highly pertinent to the focus of this study, which investigates the propagation of precipitation deficit drought and groundwater drought. The SPEI is advantageous as it considers both precipitation and evapotranspiration, providing a comprehensive view of the water balance. Moreover, its extensive application in related research further substantiates its suitability and effectiveness in our analysis [5-8].
Point 8: Some data regression is used for continuous long-series monitoring. Refer to other approximate methods also, such as DOI: 10.32604/cmes.2021.012595.
Response 8: Thanks for your kind reminder. We have studied this high-quality research and incorporated the reference [9-11] into our manuscript. (Line 195-198)
Point 9: Why is the standardized groundwater drought index given by Eq.(8)?
Response 9: The standardized groundwater drought index provided by Equation (8) is derived using the nonparametric normal quantile transformation method, which was originally applied by Bloomfiled [12]. This approach is employed to transform the percentile rankings into a normally distributed set of percentile values. By doing so, the resultant index values facilitate the comparison of drought conditions over different time periods and geographical areas, regardless of the original data distribution.
We also added the above description of SGD index before Eq 8 to make the SGD more clearly.
Point 10: Drought Duration and Drought Severity Density values are almost the same. Explain why.
Response 10: Thank you for your suggestions. Drought duration and severity are two primary attributes characterizing drought. According to the definitions, drought event duration is calculated as the time span between the occurrence and end times of a drought event, while drought severity is the cumulative absolute value of the index during the drought period. The average drought duration presented in the research is the ratio of the total drought occurrence time over 17 years to the number of drought events for each grid. The average drought severity is the ratio of the cumulative absolute value of the index over all drought periods to the number of drought events. The relationship between these two drought characteristics is as follows:
Further derivation leads to:
In this study, the minimum threshold of drought is 0.5, and the maximum threshold is 3. According to the percentile of SPEI in the drought period, the SPEI value of the 25th quantile corresponds to 0.72 and that of the 75th quantile corresponds to 1.24. Therefore, the ratio of to is mainly between 0.72 and 1.24. Numerous studies also indicate that these two drought characteristics are interrelated and exhibit a high degree of consistency. For instance, Ullah and Akbar [13] quantified the strength of drought characteristics using Kendall's tau (0.886), Spearman's rank correlation (0.969), and Pearson correlation (0.947). Gumus et al. [14] found that the Pearson correlation coefficient between hydrological drought duration and drought severity in the Orontes basin ranged from 0.92 to 0.98. Additionally, some studies, based on Copula functions, revealed the nonlinear dependence relationship between drought duration and severity [15]. Although drought characteristics and drought duration are similar, we need to note local differences and that the two types of drought characteristics change over time scales.
Point 11: The current work concentrated only on a certain domain of China. Discuss how it can be adjusted to the rest of the world. How can the results be used in the rest of the world?
Response 11: Thank you for your suggestions. we acknowledge that the current study has focused exclusively on a specific region in China. To extend the applicability of our findings to the rest of the world, several adjustments and considerations can be made. First of all, the analysis process of this study can also be applied to other research areas in the world. No matter it is based on actual measurement, remote sensing observation or model simulation, the drought characteristics and propagation process can be quantified based on the framework of this study. We focus on the transmission process of drought, including accumulation, hysteresis, delay and prolonged phenomena. This has implications for other local systems to understand the propagation laws of drought in time and space scale, and is helpful to take drought mitigation measures in advance. With the improvement of groundwater monitoring networks in various countries, large-scale groundwater drought assessment based on high density and high frequency in-situ groundwater monitoring data becomes more and more important. This work can not only provide the real situation of groundwater drought, but also calibrate hydrological models and remote sensing models. When promoting our research work, it is important to fully consider the specific environment and climate conditions of the target area, and make appropriate adjustments and customization, such as selecting appropriate drought indicators based on the characteristics of the study area, the availability of data, and the study period. In this way, our methods and findings will be more relevant and practical, providing useful references for water management and drought research around the world.
Special thanks to you for your good comments.
There is no doubt that your responsible detailed suggestions and high-quality reviews have significantly improved the quality of the article. We appreciate for your warm work earnestly, and hope that the correction will meet with approval.
Once again, thank you very much for your comments and suggestions.
- 1. Gräler, B.; E. Pebesma and G. Heuvelink. "Spatio-temporal interpolationusing gstat." R JOURNAL 8 (2016): 204-18.
- 2. Snepvangers, J. J. J. C.; G. B. M. Heuvelink and J. A. Huisman. "Soil water content interpolation using spatio-temporal kriging with external drift." Geoderma 112 (2003): 253-71. https://doi.org/10.1016/S0016-7061(02)00310-5.
- 3. Hu, H. Spatiotemporal regression kriging forestimating pm2.5 concentrationusing aerosol optical depth remotesensing data. Wuhan University, 2018, Doctor.
- 4. Bogner, K.; F. Pappenberger and H. L. Cloke. "Technical note: The normal quantile transformation and its application in a flood forecasting system." Hydrology and Earth System Sciences 16 (2012): 1085-94. 10.5194/hess-16-1085-2012.
- 5. Vicente-Serrano, S. M.; S. Beguería and J. I. López-Moreno. "A multiscalar drought index sensitive to global warming: The standardized precipitation evapotranspiration index." Journal of Climate 23 (2010): 1696-718. https://doi.org/10.1175/2009JCLI2909.1.
- 6. Cui, Y. Q.; B. Zhang; H. Huang, et al."Identification of seasonal sub-regions of the drought in the north china plain." Water 12 (2020): 10.3390/w12123447.
- 7. Wang, F.; Z. Wang; H. Yang, et al.Copula-based drought analysis using standardized precipitation evapotranspiration index: A case study in the yellow river basin, china. 11. 2019,
- 8. Zhu, Y.; Y. Liu; W. Wang, et al."A global perspective on the probability of propagation of drought: From meteorological to soil moisture." Journal of Hydrology 603 (2021): 10.1016/j.jhydrol.2021.126907.
- 9. Serrano, S. E. "A simple approach to groundwater modelling with decomposition." Hydrological Sciences Journal 58 (2013): 177-85. 10.1080/02626667.2012.745938. https://doi.org/10.1080/02626667.2012.745938.
- 10. Turkyilmazoglu, M. "Nonlinear problems via a convergence accelerated decomposition method of adomian." Computer Modeling in Engineering \& Sciences 127 (2021): 10.32604/cmes.2021.012595.
- 11. Leaf, A. T.; M. N. Fienen. "Modflow-setup: Robust automation of groundwater model construction." Frontiers in Earth Science 10 (2022): 10.3389/feart.2022.903965.
- 12. Bloomfield, J.; B. Marchant. "Analysis of groundwater drought building on the standardised precipitation index approach." Hydrology and Earth System Sciences 17 (2013): 4769-87. 10.5194/hess-17-4769-2013.
- 13. Ullah, H.; M. Akbar. "Bivariate drought risk assessment for water planning using copula function in balochistan." ENVIRONMENTAL MODELING & ASSESSMENT 28 (2023): 447-64. 10.1007/s10666-023-09880-7.
- 14. Gumus, V.; Y. Avsaroglu; O. Simsek, et al."Evaluating the duration, severity, and peak of hydrological drought using copula." Theoretical and Applied Climatology 152 (2023): 1159-74. 10.1007/s00704-023-04445-w. https://doi.org/10.1007/s00704-023-04445-w.
- 15. Li, M.; G. Wang; S. Zong, et al.Copula-based assessment and regionalization of drought risk in china. 20. 2023,

Round 2
Reviewer 2 Report
Comments and Suggestions for Authors
Accept
Reviewer 3 Report
Comments and Suggestions for Authors
Authors have improved the earliear version to make it aligned with a research paper.
Comments on the Quality of English LanguageOk.